# Rethinking Self-driving: Multi-task Knowledge for Better Generalization and Accident Explanation Ability

## Abstract

Current end-to-end deep learning driving models have two problems: (1) Poor generalization ability of unobserved driving environment when diversity of training driving dataset is limited (2) Lack of accident explanation ability when driving models don't work as expected. To tackle these two problems, rooted on the believe that knowledge of associated easy task is benificial for addressing difficult task, we proposed a new driving model which is composed of perception module for *see and think* and driving module for *behave*, and trained it with multi-task perception-related basic knowledge and driving knowledge stepwisely. Specifically segmentation map and depth map (pixel level understanding of images) were considered as *what & where* and *how far* knowledge for tackling easier driving-related perception problems before generating final control commands for difficult driving task. The results of experiments demonstrated the effectiveness of multi-task perception knowledge for better generalization and accident explanation ability. With our method the average sucess rate of finishing most difficult navigation tasks in untrained city of CoRL test surpassed current benchmark method for 15 percent in trained weather and 20 percent in untrained weathers.

## 1 Introduction

Observing progressive improvement in various fields of pattern recognition with end-to-end deep learning based methods(Krizhevsky et al., 2012; Girshick, 2015), self-driving researchers try to revolutionize autonomous car field with the help of end-to-end deep learning techniques(Bojarski et al., 2016b; Chen et al., 2015; Codevilla et al., 2017). Impressive results have been acquired by mapping camera images directly to driving control commands(Bojarski et al., 2016b) with simple structure similar to ones for image classfication task(Simonyan & Zisserman, 2014). Further researches were conducted to improve the performance of deep learning based autonomous driving system, for example, Conditional Imitation Learning(Codevilla et al., 2017) approach has been proposed to solve the ambigious action problem.

However, two crutial problems failed to be spotted: (1) Poor generalization ability of unobserved driving environment given limited diversity of training scenerios. For example, though Dosovitskiy et al. (2017) addressed the driving direction selection problem, it showed poor generalization ability in unseen test town which has different map and building structure than training town's. This generalization problem is extremely important since collected driving dataset always has limitation of diversity (2) Current end-to-end autonomous approaches lack of *accident explanation ability* when these models behave unexpectedly. Although saliency map based visualization methods(Smilkov et al., 2017; Sundararajan et al., 2017; Springenberg et al., 2014; Bojarski et al., 2016a) have been proposed to dig into the 'black box', the only information these methods could bring is the possible attention of the model instead of the perception process of the model.

We proposed a new driving approach to solve the two aforementioned problems by using multi-task basic perception knowledge. We argue that when end-to-end model is trained to address a specific difficult task, it's better to train the model with some basic knowledge to solve relevant easier tasks

before(Pan et al., 2010). An analogy for this can be observed when human beings learn a difficult knowledge. For example, to solve a complex integration problem, compared with students without basic math knowledge, students who know about basic knowledge of math are able to learn the core of intergration more quickly and solve other similar integration problems instead of memorizing the solution of the specific problem.

Our proposed model consists of two modules: perception module and driving module as in Fig. 1. The perception module is used for learning easier driving-related perception knowledge, which we refer as *ability of pixel level understanding of input* including *what & where* and *how far* knowledge. We trained perception module with segmentation map and depth map first, while the former serves as *what & where* knowledge and the latter serves as *how far* knowledge. By visualizing inferenced segmentation and depth results whether perception process works well or not could be inferred. After the perception module was trained to have ability of pixel level understanding of its image input, we freezed the perception module weights and trained driving module with driving dataset. This decomposition of end-to-end driving network strucuture is considered to be mediated perception approach(Ullman, 1980). With our proposed driving structure and stepwise training strategy, the generalization and accident explanation problems were addressed to a certain extent.

## 2 RELATED WORK

Depending on whether mediated perception knowledge are generated, self-driving models are categorized into mediated perception approach(Ullman, 1980) and behavior reflex approach.

For mediated perception approaches, there are several well-behaved deep learning methods, for example, Deep-Driving method(Chen et al., 2015) fisrtly converts input RGB images to some key perception indicators related to final driving controls. They designed a very simple driving controller based on predicted perception indicators. Problem of this approach is that key perception indicators have limitation of describing unseen scenerios and are difficult to collect in reality. Except for inferencing for final driving controls, there are approaches which focus on inferencing intermediate description of driving situation only. For separate scene understanding task, car detection(Lenz et al., 2011) and lane detection(Aly, 2008) are two main topics in this area.

Instead of inferencing one perception task at most, multi-task learning method aims at tackling several relevant tasks simultaneously. Teichmann et al. (2016) uses input image to solve both object detection and road segmentation tasks. Branched E-Net(Neven et al., 2017) not only infers for segmentation map, but also depth map of current driving scenarios. These multi-task learning methods shows better result when sharing the encoder of different perception tasks together, but they haven't really tried to make the car drive either in simulator or reality.

As for behavior reflex approach which is also called 'end-to-end learning', NVIDIA firstly proposed a model for mapping input image pixels directly to final driving control output(steer only)(Bojarski et al., 2016b). Some other approaches further atempted to create more robust models, for example, long short-term memory (LSTM) was utilized to make driving models store a memory of past(Chi & Mu, 2017).

One problem is that aforementioned methods were tested in dissimlar driving scenerios using different driving dataset, thus it's hard to determine if model itself is the source of the better driving behavior instead of effectiveness of data(Sun et al., 2017).

Codevilla et al. (2017) was tested in a public urban driving simulator Dosovitskiy et al. (2017) and sucessed to tackle the ambigous action problem which refers as optimal driving action can't be inferred from perceptual input alone. Benefit from CoRL test in Dosovitskiy et al. (2017), fair comparision could be conducted using same driving dataset. Codevilla et al. (2017) showed limitation of generalization ability problem in test town different from train town(Dosovitskiy et al., 2017) as in CoRL test training dataset could be only collected from single train town.

When the end-to-end driving method behaves badly and causes accidents, accident explanation ability is required. Though saliency-map based visualization methods(Bojarski et al., 2016a; Smilkov et al., 2017) help understand the influence of input on final driving control, it's extremely hard to derive which module of the model fails when driving problems happen — If the model percepts incorrectly or the driving inference processes wrongly based on good perception information. Driving system was enabled to give quantitative explanation by visualizing inferenced multi-task basic knowledge to solve this problem.

## 3    FRAMEWORK OF PROPOSED SYSTEM

Basic strucure of the proposed model is shown in Fig. 1. The proposed model has two parts: (1) Multi-task basic knowledge perception module (2) Driving decision branch module. The perception module is used to percept the world by inferencing dpeth map and segmentation map, which is composed of one shared encoder and two decoders for two different basic perception knowledge: (1) Segmentation decoder for generating 'what & where' information by predicting segmentation maps; (2) Depth decoder for predicting 'how far' the objects in vision are by inferencing depth maps. The perception module is aimed at extracting encoded feature map containing pixel level understanding information for driving module and qualitative explanation when proposed model doesn't work as expected by visualizing the predicted segmentation and depth maps to determine if the driving problem is caused by percept process or driving process.

The driving module enbales the model to generate driving decisions for different direction following guidances. We categorized the real world driving guidance into four types: (1) Following lane (2) Turning left (3) Going straight (4) Turning right as done in Codevilla et al. (2017). For each driving guidance direction, there is a driving branch(which predicts the value of driving controls) corresponding to it, therefore there are four driving guidance branches totally. The output of second last layer in perception module is inputted to the driving module, therefore the training of which could benefit from the multi-knowledge extracted by the perception module. Instead of linear layers, convolution layers are utilized for inferencing final driving controls for each direction, which helps keeping the spatial relation of information and reducing number of parameters as non-negligible quantity of direction branches.

### 3.1    MULTI-TASK BASIC KNOWLEDGE PERCEPTION MODULE

The perception module is built with residual block proposed in (He et al., 2016) which solves gradient vanishing and 'degradation problem', and it has a structure similar to Segnet(Badrinarayanan et al., 2015) prosposed for efficient image segmentation task. Huge difference is that in our proposed method there are two different decoders for inferencing both segmentation and depth maps simultaneously instead of segmentation map only. Besides, we constraint the total strides in encoder to 8 for keeping resolution of feature map, as large total stride has negative influence on feature map size reconstuction. Hybrid Dilated Convolution Wang et al. (2017) is adapted as last part of the encoder as it enlarges the receptive field and avoids theoretical issue of gridding problem. Groupout(Park) is also adapted to avoid overfitting problem in the convolution network.

### 3.2    DRIVING DECISION BRANCH MODULE

The driving module is built with residual block and has a general form as Codevilla et al. (2017) in last output layer for several direction outputs. It is all based on convolutional layers in order to keep the spatial information and reduce parameters motivated by Springenberg et al. (2014). Four different high level driving guidance such as "turning right" are utilized for selecting which direction branch's output is supposed to be considered as final driving outputs. Driving outputs contain steering and acceleration/brake, both of them range from -1 to 1. Since there are 4 output branches corresponding to 4 high level driving guidances, 8 feature map size convolution kernels are set in the last layer for output scalar value, in which each two are regarded as driving controls for one driving guidance. To determine the limitation of RGB image, no other information such as current speed or steering angle were used as input. Instead we atempted to predict the current speed

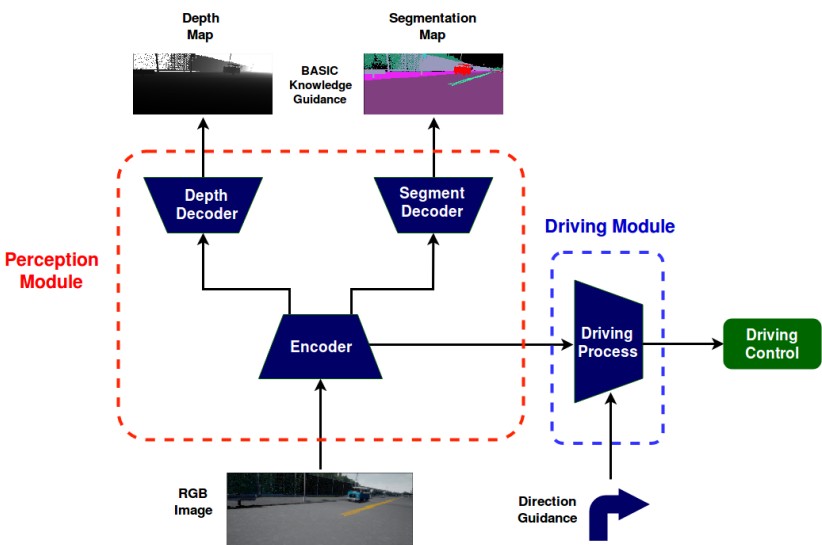

Figure 1: Basic structure of proposed self-driving system. The components which lie in the red bounding box are refered as parts of perception module. The components which lie in the blue bounding box forms the driving module. In order to see the limiation of RGB image, only current RGB image and driving guidance are used as inputs seperatedly for perception module and driving module.

based on current RGB image to keep the driving smoothly as done in (Codevilla et al., 2017). The input of the driving module is not from the last layer's output of the encoder part in the perception module, but the second last layer's output of the encoder part due to empirically selection for best generalization.

## 4 EXPERIMENTS

### 4.1 SYSTEM SETUP

The training dataset is collected in CARLA simulator(Dosovitskiy et al., 2017). CARLA simulator is a self-driving simulator developed by Intel Co. for collecting self-driving related information and evaluating driving model with a standard testing environment named CoRL test. CoRL test is composed of 4 tasks of increasing difficulty: (1) Straight: the goal is straight ahead of the starting position. (2) One turn: getting to the goal takes one turn, left or right. (3) Navigation: navigation with an arbitrary number of turns. (4) Navigation with dynamic obstacles: same as previous task, but with other vehicles and pedestrians(Dosovitskiy et al., 2017).

The main metric for quantitatively evaluating is the average success rate of finishing seperate tasks in CoRL test. CoRL test contains tests both in trained town and untrained town under both trained and untrained weathers. Test trained town and untrained town are constructed with different maps and different building texture.

### 4.2 DATASET

Dataset for training our model could be categorized into 2 items: (1) Perception module training dataset (2) Driving module training dataset. For perception module, we trained it with 35,000 pairs of RGB images, segmentation and depth maps and evaluated with 5,000 pairs. As for driving module, we trained with 455,000 dataset, and evaluated on 62,000 evaluation dataset. Before training our proposed model, two vital data processing methods were used: balancing dataset and data augmentation.

Table 1: Quantitive evaluation of methods in the goal-directed navigation tasks in CoRL test. The table reports the percentage of success rate of finishing corresponding task in different condition. Higher means better performance.

| Task | Training conditions | | | | New town | | | | New weather | | | | New town&weather | | | |
|------|----|----|----|------|----|----|----|------|-----|----|----|------|----|----|----|------|
| | MP | IL | RL | OURS | MP | IL | RL | OURS | MP | IL | RL | OURS | MP | IL | RL | OURS |
| Straight | **98** | 95 | 89 | **98** | 92 | 97 | 74 | **100** | **100** | 98 | 86 | **100** | 50 | 80 | 68 | **96** |
| One turn | 82 | **89** | 34 | 87 | 61 | 59 | 12 | **81** | **95** | 90 | 16 | 88 | 50 | 48 | 20 | **82** |
| Navigation | 80 | **86** | 14 | 81 | 24 | 40 | 3 | **72** | **94** | 84 | 2 | 88 | 47 | 44 | 6 | **78** |
| Nav. dynamic | 77 | **83** | 7 | 81 | 24 | 38 | 2 | **53** | **89** | 82 | 2 | 80 | 44 | 42 | 4 | **62** |

For fair comparison, we use same driving dataset published by Conditional Imitation Learning(Codevilla et al., 2017) except that we collected extra segmentation and depth maps in train town for training our proposed perception module.

### 4.2.1 DATA BALANCING

Dataset balancing contributed to better generalization of both perception module and driving module in our experiments as it enables each mini-batch to be a microcosm of the whole dataset. For perception module, dataset were balanced to ensure that each mini-batch contains all different training weathers and an equal amount of going straight and turning situations. For driving module, we balance each training mini-batch to ensure equally distribution of different driving direction guidance, and reorganized large steer(absolute value larger than 0.4) data accounts for 1/3 in each mini-batch, brake situation data acounts for 1/3, noise steer situation for 1/10.

### 4.2.2 DATA AUGMENTATION

We add guassian noise, coarse dropout, contrast normalization, Guassian blur to both training dataset for perception and driving module for enlarging training dataset distribution.

### 4.3 TRAINING DETAILS

We trained the whole system using a step-wise training method which is firstly we trained the perception module with multi-task basic perception knowledge, then we freezed the weights of perception module and train driving module with driving dataset. For training the perception module, we used mini-batch size 24 and set ratio of segmentation loss and depth loss to be 1.5:1. Softmax categorical crossentropy is used for segmentation loss, binary crossentropy is used for depth loss. Adam of 0.001, which is multiplied by a factor of 0.2 of previous learning rate if validation loss does't drop for 1 epoch is used as optmizer. L2 weight decay and early stopping are also used for avoid overfitting. As for training the driving module, we consider MSE loss and use Adam with starting learning rate of 0.002 which exponentially decay of 0.9 every epoch. Early stopping and L2 decay are used for regularization.

## 5 EXPERIMENTS & RESULTS

### 5.1 GENERALIZATION ABILITY TEST

We compare the results of driving performance between our proposal and other methods tested in CoRL test via success rate of finishing each task. The details of results are shown in Table. 1

From Table. 1, though our proposal finished slightly less in training conditions comparing with other methods, our proposal achieved much higher success rate in untrained town environments, which demonstrates our model has much better generalization ability of adapting to untrain town than other methods tested in the CoRL test when trained with limited diversity of training conditions. One important notice is that we use the almost the same driving dataset for training as the method Codevilla et al. (2017) showed in the Table. **??**.

We could also visualize the perception process when the model works. One example of test in untrained town and untrained weather is shown in Fig. 2.

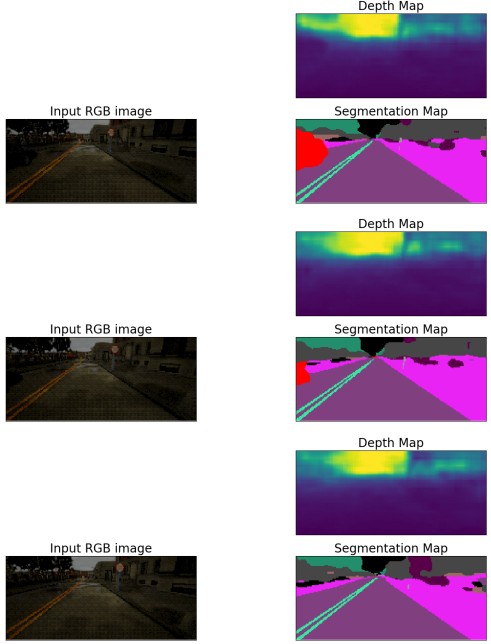

Figure 2: Screenshots of captured single RGB image and our proposal's inference results which consists of predicted segmentation map and depth map during test in untrained town under untrained weather. Obviously from the inferenced segmentation and depth maps we get information that the driving model knows a car is passing by the left side.

## 5.2 Origin of Better Gerneralization Ability

Since we observed our training model has better generalization ability in unseen town comparing with other methods when almost the same driving dataset were used to train (except that we collected extra depth maps and segmentation maps in same training environments), we want to investigate the origin of the better generalization ability. There are two possible reasons why our training model has better generalization ability in unseen town: (1) Basic knowledge (segmentation map and depth map) (2) Network structure. Therefore we conduct experiments by comparing performance of two methods:

- Our original proposal: Firstly train perception module with basic knowledge, after training perception module, freeze its weights and train driving module with driving dataset
- Compared method: Train the encoder of perception module and driving module together with driving dataset. No basic perception knowledge is used for training model.

Since tests in CoRL cost much time, we limited our evaluation to the most difficult untrained town under untrained weathers test. Results are shown in Table. 2. From the results it's obvious that multi-basic knowledge we use in the training phase is the origin of our proposal's good generalization ability of untrained town instead of the network structure. Moreover, the network structure could be improved to achieve better performance in the furture.

## 5.3 Qualitative Cause Explanation Ability of Driving Problems

Besides basic knowledge leads to better generalization ability, it could also be used to give a qualitative explanation of driving problems. Basic knowledge of segmentation map and depth map

Table 2: Quantitive evaluation of original proposal which uses segmentataion and depth maps in training phase and compared method which doesn't use. Results shows that when multi-knowldge is not used for training, the success rate drops hugely in the testing town under untained weathers. It indicates that multi-basic knowledge is the main origin of the good generalization ability in our proposal.

|  | New town&weather test | |
| Task | COMPARED | ORIGINAL |
| --- | --- | --- |
| Straight | 91 | **96** |
| One turn | 52 | **82** |
| Navigation | 20 | **78** |
| Nav. dynamic | 16 | **62** |

are output from the perception module during test phase, therefore how the driving module percepts the current scenario could be known by simply visualizing the outputs of segmentation and depth from perception module. Depending on the predicted pixel understanding of the situation, cause of driving problem could be inferred.

One example is shown in Fig. 3. For a failed straight task in untrained town under untrained weather soft rain sunset as the driving model failed to move forward, we visualized outputs of segmentation and depth maps predicted by perception module. It's obvious that this failure case is caused by the perception module since the model falsely percepted that there is a car in front of it and in order to avoid collision it did't start. There is no car actually thus the perception module made false judgement. However, what's interesting is that sometimes it fools readers to think that there is a car in Fig. 3 because of sun ray reflection on the wet road and the perception module has the similar understanding as these readers. Therefore in some aspects the perception module makes the right judgement instead of wrong's. For traditional end-to-end learning driving methods(Bojarski et al., 2016b) it's impossible to reason as they don't focus on the cause explanation ability which is of great importance for practice use of deep learning driving models.

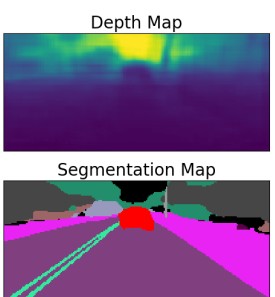

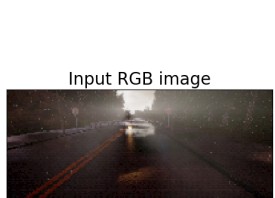

Figure 3: Failed straight task in untrained town under untrained soft rain sunset weather. From predicted segmentation and depth maps, we know that the driving model thinks that there is a car in front of it but actually there isn't any car in front of it.

### 5.4 FINE-TUNE TEST

Fine-tuneYosinski et al. (2014), which refers to use other well-trained models weights on different target-related dataset as initial weights for training with target dataset instead of using weights initializing methods(Glorot & Bengio, 2010; He et al., 2015; LeCun et al., 2012), is a common trick used in deep learning since empirically it could leads to better generalization on new target dataset. Here in our specific case we refer fine-tune method to be after training perception module we train the weights of the encoder of the perception module as well instead of freezing these weights. In

Table. 3 we compare the performance of fine-tune method and our original proposed method.

Table 3: Quantitive evaluation of original proposal and fine-tune method which doens't freeze the perception module's weights during training the driving module. Results shows that the success rate of the fine-tune method drops dramaticaly in the testing town under untained weathers.

| Task | New town&weather test | |
| --- | --- | --- |
| | FINETUNE | ORIGINAL |
| Straight | 88 | **96** |
| One turn | 59 | **82** |
| Navigation | 42 | **78** |
| Nav. dynamic | 33 | **62** |

In this comparison we achieved counter-intuition results: after fine-tune the weights of the perception module the driving model achieved worse results than original method which freeze the weights of perception module when training the driving module.

One possible reason is that the generalization ability lies in the perception module instead of the driving module, therefore when we train the perception module again with driving dataset, the ability of generating compressed multi-knowdege information is destoryed. As the fine-tune model couldn't benefit from the multi-task knowledge anymore, it failed to produce the same generalization ability as the original proposal did.

Furthermore we conduct experiment on visualizing one direction of loss surface by projecting the loss surface to 2 dimension(Goodfellow & Vinyals, 2014) to investigate some qualitative explanation for this comparison result. $x$ axis corresponds to linear interpolation of the weights of original proposed method and weights of compared fine-tuned method after training. Formulation of calculating the weights in this projection direction is Equation. 1.

$$\alpha \in [-1, 2], f(\alpha x_{finetune} + (1 - \alpha)x_{rgb0}) \tag{1}$$

$\alpha$ is linear interpolation ratio, $x_{fintune}$ and $x_{rgb0}$ are trained weights of fine-tune method and original proposal method. $f(x)$ is loss function of the whole model while input is considered as different model weights. We draw out the projected loss surface as Fig. 4 by sampling from the interpolation weigthts. From Fig. 4 we can get one possible qualitative reason for worse behavior of fine-tune method from a loss surface perspective: Model weight got by using fine-tune method is stuck in a super flat surface, while model weights of original proposed method successfully finds a local minimum.

## 6 CONCLUSION

In this paper we propose a new driving system for better generalization and accident explanation ability by enabling it to do simpler driving-related perception task before generating commands for diffult driving task. Through multiple experiments we empirically proved the effectiveness of the multi basic perception knowledge for better generalization ability of unobserved town when diversity of training dataset is limited. Besides our proposed model has self-explanation ability by visualizing the predicted segmentation and depth maps from the perception module to determine the cause of driving problems when they happen. One interesting result we acquired by comparing different train strategies is that the generalization ability of driving origins from basic knowledge and lies in weights of the perception module which should not be modified during training with driving dataset. We hope our work could movitivate other researches to use multi-task target related perception knowledge for better performance in robot learning. In future we will investigate more effective network structures.

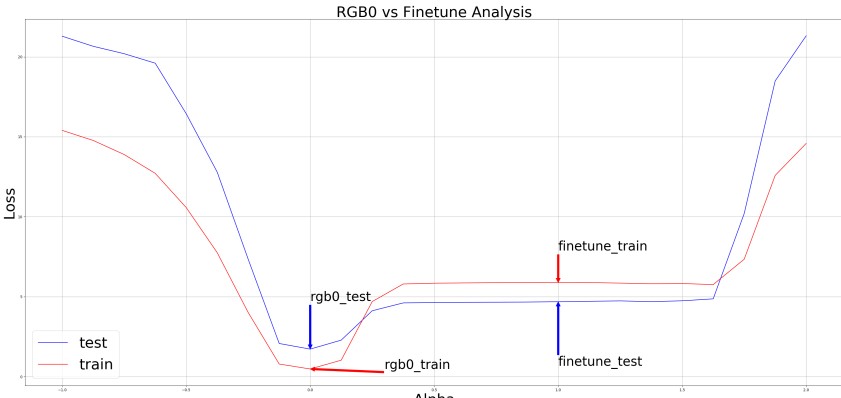

Figure 4: One perceptive of loss surface by linear interpolation of original proposed method weights and fine-tune method weights. Blue line refers to test loss, red line refers to train loss. From the visualization perspective it's possibly that finetune method weights are stuck in a flat surface while the original proposed weights sucessfully find a local minimum.

## ACKNOWLEDGMENTS

Thanks to all Prof.Ogata lab members especially Kamuza SASAKI san who teaches me about Deep Learning patiently when I have zero knowledge of what it is. Great thanks to my bros Zehai TU and Pengfei LI who support me no matter how annoying I am in the midnight. Final thanks to my homie Mengcheng SONG for being a Hiphop guide for me and makes me understand about the importance of always 'keep it real'.

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
