# OpenReview forum: "RETHINKING SELF-DRIVING : MULTI -TASK KNOWLEDGE FOR BETTER GENERALIZATION AND ACCIDENT EXPLANATION ABILITY"
_ICLR.cc/2019/Conference_

### Official Review · AnonReviewer1 · 2018-10-31
**End-to-end driving with perceptual auxiliary tasks similar to Xu et al CVPR'17**

**Rating:** 3
**Confidence:** 4

**Review:**

# Summary

This submission proposes a multi-task convolutional neural network architecture for end-to-end driving (going from an RGB image to controls) evaluated using the CARLA open source simulator. The architecture consists of an encoder and three decoders on top: two for perception (depth prediction and semantic segmentation), and one for driving controls prediction. The network is trained in a two-step supervised fashion: first training the encoder and perception decoders (using depth and semantic segmentation ground truth), second freezing the encoder and training the driving module (imitation learning on demonstrations). The network is evaluated on the standard CARLA benchmark showing better generalization performance in new driving conditions (town and weather) compared to the CARLA baselines (modular pipeline, imitation learning, RL). Qualitative results also show that failure modes are easier to interpret by looking at predicted depth maps and semantic segmentation results.


# Strengths

Simplicity of the approach: the overall architecture described above is simple (cf. Figure 1), combining the benefits of the modular and end-to-end approaches into a feed-forward CNN. The aforementioned two-stage learning algorithm is also explained clearly. Predicted depth maps and semantic segmentation results are indeed more interpretable than attention maps (as traditionally used in end-to-end driving).

Evaluation of the driving policy: the evaluation is done with actual navigation tasks using the CARLA (CoRL'18) benchmark, instead of just off-line behavior cloning accuracy (often used in end-to-end driving papers, easier to overfit to, not guaranteed to transfer to actual driving).

Simple ablative analysis: Table 2 quantifies the generalization performance benefits of pretraining and freezing the encoder on perception tasks (esp. going from 16% to 62% of completed episodes in the new town and weather dynamic navigation scenario).


# Weaknesses

## Writing

I have to start with the most obvious one. The paper is littered with typos and grammatical errors (way too many to list). For instance, the usage of "the" and "a" is almost non-existent. Overall, the paper is really hard to read and needs a thorough pass of proof-reading and editing. Also, please remove the acknowledgments section: I think it is borderline breaking the double-blind submission policy (I don't know these persons, but if I did that would be a breach of ICLR submission policy). Furthermore, I think its contents are not very professional for a submission at a top international academic venue, but that is just my opinion.


## Novelty

This is the main weakness for me. The architecture is very close to at least the following works:
- Xu, H., Gao, Y., Yu, F. and Darrell, T., End-to-end learning of driving models from large-scale video datasets (CVPR'17): this reference is missing from the paper, whereas it is very closely related, as it also shows the benefit of a segmentation decoder on top of a shared encoder for end-to-end driving (calling it privileged training);
- Codevilla et al's Conditional Imitation Learning (ICRA'18): the only novelty in the current submission w.r.t. CIL is the addition of the depth and segmentation decoders;
- Müller, M., Dosovitskiy, A., Ghanem, B., & Koltun, V., Driving Policy Transfer via Modularity and Abstraction (CoRL'18): the architecture also uses a shared perception module and segmentation (although in a mediated way instead of auxiliary task) to show better generalization performance (including from sim to real).

Additional missing related works include:
- Kim, J. and Canny, J.F., Interpretable Learning for Self-Driving Cars by Visualizing Causal Attention (ICCV'17): uses post-hoc attention interpretation of "black box" end-to-end networks;
- Sauer, A., Savinov, N. and Geiger, A., Conditional Affordance Learning for Driving in Urban Environments (CoRL'18): also uses a perception module in the middle of the CIL network showing better generalization performance in CARLA (although a bit lower than the results in the current submission).
- Pomerleau, D.A., Alvinn: An autonomous land vehicle in a neural network (NIPS'89): the landmark paper for end-to-end driving with neural networks!


## Insights / significance

In light of the aforementioned prior art, I believe the claims are correct but already reported in other publications in the community (cf. references above). In particular, the proposed approach uses a lot more strongly labeled data (depth and semantic segmentation supervision in a dataset of 40,000 images) than the competing approaches mentioned above. For instance, the modular pipeline in the original CARLA paper uses only 2,500 labeled images, and I am sure its performance would be vastly improved with 40,000 images, but this is not evaluated, hence the comparison in Table 1 being unfair in my opinion. This matters because the encoder in the proposed method is frozen after training on the perception tasks, and the main point of the experiments is to convince that it results in a great (fixed) intermediate representation, which is in line with the aforementioned works doing mediated perception for driving.

The fine-tuning experiments are also confirming what is know in the litterature, namely that simple fine-tuning can lead to catastrophic forgetting (Table 3).

Finally, the qualitative evaluation of failure cases (5.3) leads to a trivial conclusion: a modular approach is indeed more interpretable than an end-to-end one. This is actually by design and the main advocated benefit of modular approaches: failure in the downstream perception module yields failure in the upstream driving module that builds on top of it. As the perception module is, by design, outputting a human interpretable representation (e.g., a semantic segmentation map), then this leads to better interpretation overall.


## Reproducibility

There are not enough details in section 3.1 about the deep net architecture to enable re-implementation ("structure similar to SegNet", no detailed description of the number of layers, non-linearities, number of channels, etc).

Will the authors release the perception training dataset collected in CARLA described in Section 4.2?



# Recommendation

Although the results of the proposed multi-task network on the CARLA driving benchmark are good, it is probably due to using almost two orders of magnitude more labeled data for semantic segmentation and depth prediction than prior works (which is only practical because the experiments are done in simulation). Prior work has confirmed that combining perception tasks like semantic segmentation with end-to-end driving networks yield better performance, including using a strongly related approach (Xu et al). In addition to the lack of novelty or new insights, the writing needs serious attention.

For these reasons, I believe this paper is not suitable for publication at ICLR.

---

### Official Review · AnonReviewer3 · 2018-11-02
**The paper is not bad technically, but the contributions is not good enough**

**Rating:** 4
**Confidence:** 5

**Review:**

Major Contribution:
This paper details a method for a modified end-to-end architecture that has better generalization and explanation ability. The paper outlines a method for this, implemented using an autoencoder for an efficient feature extractor. By first training an autoencoder to ensure the encoder captures enough depth and segmentation information and then using the processed information as a more useful and compressed new input to train a regression model. The author claimed that this model is more robust to a different testing setting and by observing the output of the decoder, it can help us debug the model when it makes a wrong prediction.

Organization/Style:
The paper is well written, organized, and clear on most points. A few minor points:
1) On page 5, the last sentence, there is a missing table number.
2) I don't think the last part FINE-TUNE Test is necessary since there are no formal proofs and only speculations.

Technical Accuracy:
The problem that the paper is trying to address is the black-box problem in the end-to-end self-driving system.
The paper proposes a method by constructing a depth image and a segmentation mask autoencoder. Though it has been proved that it is effective in making the right prediction and demonstrated that it has the cause explanation ability for possible prediction failures. I have a few points:
The idea makes sense and the model will always perform better when the given input captures more relevant and saturated representations. The paper listed two important features: depth information and segmentation information. But there are other important features that are missing. In other words, when the decoder performs bad, it means the encoder doesn't capture the good depth and segmentation features, then it will be highly possible that the model performs badly as well. However, when the model performs bad, it does not necessarily mean the decoder will perform badly since there might be other information missing, for example, failure to detect the object, lines and traffic lights etc.

In conclusion, the question is really how to get a good representation of a self-driving scene. I don't think to design two simple autoencoders for depth image construction and image segmentation is enough. It works apparently but it is not good enough.

Adequacy of Citations:
Good coverage of literature in self-driving.

---

### Official Review · AnonReviewer2 · 2018-11-13
**The paper showed the benefit of the proposed multi-task architecture, but not novel enough for ICLR level**

**Rating:** 4
**Confidence:** 4

**Review:**

This paper presents one end-to-end multi-task learning architecture for depth & segmentation map estimation and the driving prediction. The whole architecture is composed of two components, the first one is the  perception module (segmentation and depth map inference), the second one is the driving decision module. The training process is sequential, initially train the perception module, then train the driving decision task with freezing the weights of the perception module. The author evaluated the proposed approach on one simulated dataset, Experimental results demonstrated the advantage of multi-task compared to the single task.

Advantages:
The pipeline is also easy to understand, it is simple and efficient based on the provided results.
The proposed framework aims to give better understanding of the application of deep learning in self-driving car project. Such as the analysis and illustration in Figure 3.

Questions:
There are several typos needed to be addressed. E.g, the question mark in Fig index of section 5.1. There should be comma in the second sentence at the last paragraph of section 5.2.
Multi-task, especially the segmentation part is not novel for self-driving car prediction, such as Xu et al. CVPR’ 17 paper from Berkeley. The experiment for generalization shows the potential advancement, however, it is less convincing with the limited size of the evaluation data, The authors discussed about how to analyze the failure causes, however, if the perception  learning model does not work well, then it would be hard to analyze the reason of incorrectly prediction.

In general, the paper has the merits and these investigations may be helpful for this problem, but it is not good enough for ICLR.

---

### Public Comment · (anonymous) · 2018-10-02
**Simple, effective and easy to catch**

It is a simple and effective method, but it seems that the performance gain is mainly due to the extra segmentation and depth map annotations.

My confusions are as followings:
1. in terms of the settings of fine-tune method, where the encoder is not fixed during the training of driving module, did you try fine-tuning the whole model after each module is trained in the stepwise mode as the paper stated, which I think is a more reasonable fine-tuning setting?
2. in section 4.3, the claim 'binary crossentropy is used for depth loss' seems to be a typo...

---

> ### Author Response · Authors · 2018-10-03
> **Confusion explanation**
>
> Firstly, thanks for your comments.  I agree with your comment that performance gain is mainly due to the extra segmentation and depth map annotations, which we refer as 'multi-task knowledge' in the manuscript.
>
> As for your confusion:
> 1. Perception training dataset and driving training dataset were not collected simultaneously, which means we do not have 'inpput RGB image, segmentation map, depth map, driving controls' pairs for training driving module. There are two reasons for this: firstly there are lots of published real driving dataset which consists of RGB image and driving commands which could be reused, but only little of them have corresponding segmentation and depth maps. Secondly we want to focus on the effectiveness of 'multi-task knowledge' instead of 'new combination of driving training dataset'.
>
> We tried to finetune the whole model after each module was trained with iterative training with 'perception dataset' and 'driving dataset',  however we don't get better results.
>
> 2. We used binary crossentropy for depth loss, as we normalized depth loss from 0-1 and considered it as a two category classification when training and found that worked better than other loss like 'MSE' .

---

### Meta-Review · Area_Chair1 · 2018-12-05
**Simple design to address generalizability and interpretability, but needs more work**

**Confidence:** 4
**Recommendation:** Reject

**Metareview:**

The paper presents a unified system for perception and control that is trained in a step-wise fashion, with visual decoders to inspect scene parsing and understanding. Results demonstrate improved performance under certain conditions. But reviewers raise several concerns that must be addressed before the work is accepted.

Reviewer Pros:
+ simple elegant design, easy to understand
+ provides some insight behind system function during failure conditions (error in perception vs control)
+ improves performance under a subset of tested conditions

Reviewer Cons:
- Concern about lack of novelty
- Evaluation is limited in scope
- References incomplete
- Missing implementation details, hard to reproduce
- Paper still contains many writing errors